# Shifting the gaze of the physician from the body to the body in a place: A qualitative analysis of a community-based photovoice approach to teaching place-health concepts to medical students

Lauri Andress[1]*, Matthew P. Purtill[2]

1 Department of Health Policy, Management & Leadership, School of Public Health, West Virginia University, Morgantown, West Virginia, United States of America, 2 Department of Geology & Environmental Sciences, State University of New York at Fredonia, Fredonia, New York, United States of America

* laandress@hsc.wvu.edu

**Data Availability Statement:** All relevant data are within the manuscript and its Supporting Information files.

## Abstract

Medical practitioners, trained to isolate health within and upon the body of the individual, are now challenged to negotiate research and population health theories that link health status to geographic location as evidence suggests a connection between place and health. This paper builds an integrated place-health model and structural competency analytical framework with nine domains and four levels of proficiency that is utilized to assess a community-based photovoice project's ability to shift the practice of medicine by medical students from the surface of the body to the body within a place. Analysis of the medical student's photovoice data demonstrated that the students achieved structural competency level 1 proficiency and came to understand how health might be connected to place represented by six of the nine domains of the structural competency framework. Results suggest that medical student's engagement with place-health systemic, institutional and structural forces deepens when they co-create narratives of their lived experiences in a place with patients as community members during a community-based photovoice project. Given the importance of place-health theories to explain population health outcomes, a place-health model and structural competency analytical framework utilized during a community-based photovoice project could help medical students merge the image of patients as singular bodies into bodies set within a context.

## Introduction

Medical practitioners, trained to isolate health within and upon the body of the individual, are now challenged with understanding and integrating research and population health theories that link health status to place. This study provides an example of different models, theories, and pedagogies that may be used to teach medical students how to expand the concept of

**Funding:** Authors received no specific funding for this work.

**Competing interests:** The authors have declared that no competing interests exist.

health beyond the body to take account of place and the structures, systems, and institutions where the bodies they treat as patients are located. The paper reports on the development and application of an integrated place-health model and structural competency analytical (SCA) framework to assess a community-based photovoice project who's goal was to heighten the ability of medical students to understand the health risks associated with places and to shift the practice of medicine from the surface of the body to the body within a place.

## Place-health concepts

Places where people reside hold the key to the state of health and disease manifest within and on bodies. However, connecting downstream health problems (e.g., chronic diseases) to upstream issues and characteristics of communities is challenging for medical models, pedagogy, and tools [1, 2] (Fig 1). Consequently physicians are rarely prepared to consider the social and systemic issues of a place because health models and medical training and practices either omit or fail to integrate place-health concepts that could explicate the structural, historical, and institutional features of places where patients reside [1–8].

In her book, *Urban Alchemy*, Mindy Fullilove, a pioneer in psychiatric public health, used systems therapy to demonstrate how population health epidemics, e.g., HIV, drug abuse and addiction, tuberculosis, are best understood as symptoms of disorder within the mechanisms of a place [7, 9]. For Fullilove, the systems approach required an understanding of how cities, neighborhoods, and regions are designed, socialized, and managed. The presence of these downstream health problems like drug abuse and tuberculosis suggested to Fullilove that causation and solutions to health problems go beyond individual bodies defined as patients to bodies defined as social beings impacted by a place with systemic institutional, social, and structural features [1, 4]. In fact, a foundational premise of place -health research is described as embodiment where the external physical and social worlds are taken in and expressed in human biology [10–16].

Human geography theories add a two-way dimension to embodiment with theories on the social construction of place. Bodies that have been imprinted with external physical and social worlds were firstly groups engaged in the social construction of places where they embellished a space with remembrances, imaginings, social relations, and the public assimilation of shared narratives [17–19]. Fig 1 demonstrates how the two concepts, the social construction of places by groups and embodiment, work together to configure a place and how the construction of that place distributes opportunities in relation to socioeconomic status, race, and ethnicity [16, 20–29].

The place-health model in Fig 1 is meant to resemble a pathway diagram. The model should be read from top to bottom with the least, non-health related, upstream causes of health inequities at the top flowing downstream to the more traditional factors that ultimately cause health inequities at the bottom of the model. According to the place-health model in Fig 1, inequities in health depicted at the bottom of the model originate at the top with the cultural toolkit that is attached to a place or region [30]. The toolkit is where the construction of place begins as social processes assign meaning to phenomena using a shared set of norms, system of beliefs, narratives, and distinctive spiritual, material, intellectual, and emotional features [31–34]. It is through the public assimilation of shared narratives in any given society that groups begin to convert spaces into places [35–37].

In Fig 1, acting through the cultural toolkit, groups ascribe meaning and a shared narrative to phenomena which is then used to assign worth to the social status of others. The cultural toolkit and the various meanings it assigns become the foundation upon which institutions, systems, policies and regulations are built in a place. These same institutions and systems are

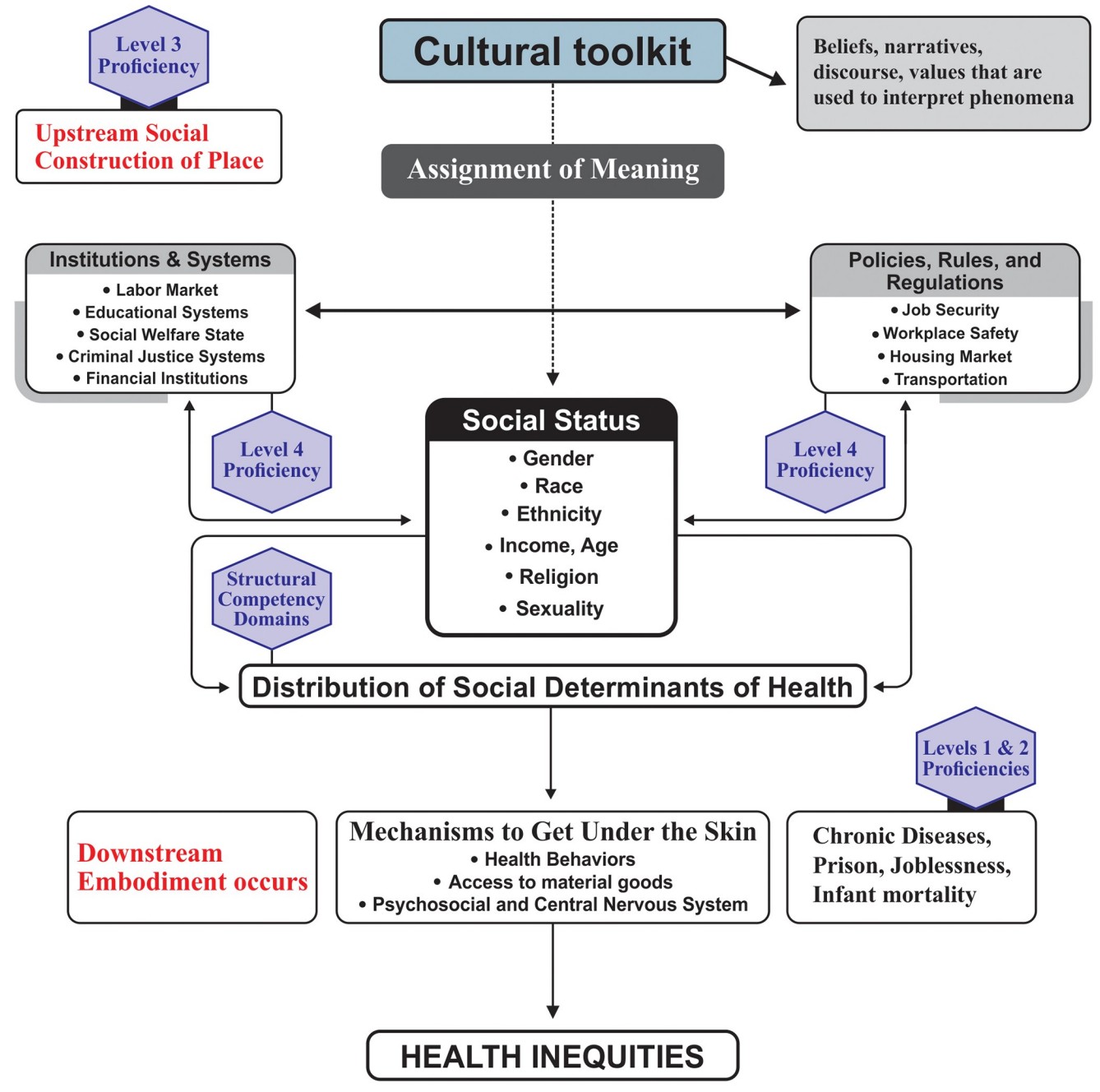

SOURCE Lead author's analysis of how place and health interact to create health inequities.

**Fig 1. A place-health model on the production of inequities.**

then used to sanction the distribution of societal opportunities and resources also referred to as the social determinants of health (SDOH). These mutually reinforcing arenas including are the SDOH in a given region that serve to regulate the place in which the citizens live [38, 39].

Finally, the bottom of the model demonstrates the concept of embodiment when it portrays how the SDOH, (i.e., experiences, opportunities, and resources) of a place get under the skin of marginalized groups by three routes: (1) influencing behavior and making behavior harmful; (2) restricting the distribution of and access to key resources; and/or (3) by causing deep seated, chronic anxiety and physiological stress [24, 30, 40–43].

A final principle of the place-health model are the Upstream and Downstream designations to the left of the diagram noting that a society can intentionally use its collective autonomy to work in any or all of these areas: 1) downstream, where illness already exist; 2) upstream on the structural issues including policies, systems, or regulations); or 3) furthest upstream on the cultural toolkit to explore and change the socially constructed, values, beliefs, and narratives that drive the assignment of meaning to group differences and other phenomena [2].

**Structural competency analytical framework.** To provide additional illustration of mechanisms that explain the association between places and health, a modified SCA framework is overlaid on and integrated into the place -health model of Fig 1. For a description of how the structural competency framework was adapted from Bourgois et al. (2017) see S3 File [44]. In comparison to cultural competency, which situates customs and individual level symptoms on the bodies of marginalized groups, structural competency extends the diagnosis of individual health and culture populations and how the features of a place including its' institutions, systems, policies, and markets shape the lives and health of those groups [1, 45, 46]. Structural competency involves the recognition of structural and systemic issues as risk factors that contravene the production of health and individual clinical interventions [1, 4, 44, 45].

Nine domains and key assessment criteria of the SCA framework are defined in Table 1. Each domain represents a SDOH and pathway through which specific societal resources, systems, and social processes of a place aggravate individual patients' health and shape the health of groups [44]. All the domains are external social and institutional systems constructed by groups that become embodied. The nine domains and their embodiment are situated below the cultural toolkit which is located at the top of the model where the assignment of meaning occurs in the place—health model (Fig 1).

The SCA framework (SCA) framework is given depth by adding four levels of proficiency (Table 2). In the modified framework at level 1 proficiency the practitioner avoids a narrow diagnosis of a health problem by connecting the health problem to knowledge about how the

**Table 1. Structural competency domains.**

| Domain | Definition |
|---|---|
| Financial security | Resources to live comfortably |
| Residence | A safe, clean/private/quiet/ stable place to sleep and store possessions |
| Risk environments | Places where you spend your time each day feel safe and healthy |
| Food access | Adequate nutrition and access to healthy food |
| Social network | Social network Friends, family, or other people who help you when you need it |
| Legal status | Legal status and or legal problems |
| Education | Reading skills, language, level of education, and knowledge about the educational system |
| Structural Stigmatization | Societal-level conditions, cultural norms, and institutional policies that constrain the opportunities, resources, and wellbeing of the stigmatized. |
| Discrimination | Able to identify harm or loss of opportunities that could result from a system or institution based on a structural stigma, stereotypical biases, or negative moral judgments. |

Definitions of the nine domains of the structural competency analytical framework.

**Table 2. Structural competency levels of proficiency.**

| Level of Proficiency | Definition |
|---|---|
| 1st Level of Proficiency | Knowledge about patient that exceeds the individual body to include an understanding of how social and structural systems -the nine domains -of a place shape population health. |
| 2nd Level of Proficiency | Knowledge of external non-medical resources, practices, or policies in the community that address structural issues from the nine domains that contravene the ability of health care practices to improve well-being. |
| 3rd Level of Proficiency | Able to recognize how "I see" that patient and understand how that characterization (individual stigmatization) may be multiplied in systems to result in societal-level, structural stigmatization. |
| 4th Level of Proficiency | Acts as an informed citizen to undo unsuccessful policies, regulations, structures and systems that influence the population health of groups in a place. |

The four levels of proficiency for a practitioner from the structural competency analytical framework.

place where the patient resides contributes to the health problem seen upon the body. Next, at level 2 proficiency the medical practitioner identifies community policy or systems resources that could help the patient if the external problem, acting as one of the nine domains, contravenes the ability of the patient to be healthy and/or adhere to healthcare interventions. At levels 1 and 2 the practitioner is not expected to understand how places are socially constructed within the cultural toolkit. Rather the practitioner will simply understand that external factors shape health. In the place- health model proficiency levels 1 and 2 occur near the distribution of the SDOH also described as the structural competency domains (Fig 1). It is possible for a practitioner to comprehend that these domains shape health but not question or wonder why some groups experience these circumstances more than others.

In the place- health model, the vision for proficiency at levels 3 and 4 occur at the top starting at the assignment of meaning, then move into the cultural toolkit followed by work among the policy sectors, institutions, and systems (Fig 1). At level 3 proficiency, located at the top of the health-place model, practitioners begin to consider the origins of what they know about groups in a place and how individual level biases translate into structural stigma. This work occurs within the cultural toolkit with discourse and the construction of narratives that explore the meaning of group differences which have been used to justify the distribution of societal resources [46, 47]. It is at the point of level 3 proficiency that the practitioner should exemplify some understanding of hierarchies and broader sets of power relationships. Acknowledgement of socially constructed places by the practitioner should demonstrate a connection between the visible, material elements of place and the unacknowledged biases, prejudices and inequalities of actors, institutions and systems that shape a space.

Level 4 proficiency is located around the institutions and polices of the place-health model. At level 4 proficiency practitioners transform into civic members actively contributing to deliberations about the systems, policies, and institutions that play a role in shaping the physical, social, and economic aspects of places that get under the skin to produce population health [18]. Political efficacy becomes important as well as the practitioner's ability to merge medical skills with the self as a civic and political being [48, 49].

## Methods

### Approach

The Institutional Review Board of West Virginia University approved the original study and subsequent analysis of data collected between March and April 2016. All study participants were informed about and agreed via written consent that at the close of the project their

photos, captions, written narratives, and interviews would be curated into an exhibition and video (see video link in S1 File). A mixed-methods study design was used to implement the original participatory photovoice project that included data from medical students and community members.

The initial project participants including medical students and community members (n = 36) were recruited using verbal invitations, fliers, and word-of-mouth among community leaders, families in the community, and medical school faculty. Qualitative data (coded interviews, photos, captions, and reflections) were gathered while demographic, household, economic, neighborhood, educational, and city infrastructure data were plotted and mapped using geographic information systems (GIS) software. The final project included an exhibition and video, "Imagining the West Side: Constructing Health through the Built Environment". The GIS maps, depictions from the exhibitions, and video may be found in S1 File. The basis for this paper is the analysis of the qualitative photovoice data from the medical students including coded interviews, photos, captions and reflections.

## Context

This project took place in Charleston, West Virginia. Specifically, research focused on the West Side neighborhood. GIS maps of all data describing the demographic profile of the community are available in S1 File. Nine percent (4,779) of Charleston's 50,911 residents live in various communities that make up the West Side with demographic data indicating a substantial African-American and mixed-race population (27.4% and 18%, respectively), and a slightly higher ratio of women to men (53% to 47%, respectively). Income disparity has strong geographic expressions. West Side populations have a high percentage of people living below the poverty line (36.6%), especially when compared to Charleston as a whole (18.7%) or West Virginia (18.0%). In 2014 inflation- adjusted dollars, West Side citizens have an average median income of $23,197.14 which is $31,993.16 below Charleston proper ($55,190.30) and $3,824.86 lower than the state median income ($27,022). Charleston Police Department 2015 figures indicate high violent and non- violent crime rates for the West District, which subsumes the West Side.

## Recruitment of medical student study participants

Medical student project participants (n = 10) were recruited using verbal invitations, fliers, and word-of-mouth among medical school faculty. The medical students had to be enrolled in the West Virginia University- School of Medicine, Charleston branch (WVU-SOM-Charleston). Medical students understood the project as an opportunity to receive an overview of the impact of community factors on health and the perceptions of citizens from the community. Of the ten medical students, seven were men and three were women. Three students were fourth year students, while the remainder where third year students.

Visual inspection by the researchers recorded that of the ten medical students, none were of Hispanic or African American origin, while three were of another ethnicity. In an email WVU -SOM administrators confirmed that all of the students identified as white, while of those ten, three were members of an Asian ethnic group.

Medical school participants partnered with West Side community leaders and members. During this effort community members and medical students shared stories about their lives and where they grew up, learned about the SDOH, and participated in a neighborhood tour narrated by community members.

All study participants were informed about and agreed via written consent that at the close of the project their photos and interviews would be curated into an exhibition and video (see video link in S1 File).

## Community-based photovoice research

The use of a community-based photovoice process allowed medical students to acquire knowledge about the place, demonstrate how the knowledge was incorporated into their role as a medical practitioner and convey the confluence of these experiences through interviews, written reflections and photos. Community-based participatory research (CBPR) recognizes that external parties to a community are better informed when they work in partnership with community members [50, 51]. A detailed description of the community sessions in this study may be found in S2 File.

Photovoice, considered a form of CBPR, facilitates co-creation and opens up a conversation where community members and external agents may explore person-place dynamics, document contextual strengths, assess complex conditions, and examine policies that influence the lives of the community [52–55]. Also considered a visual storytelling method, photovoice facilitates exploration of complex conditions including the lived experiences of those who are influenced by institutions, systems, programs, and policies of a place [17–19].

For this study, the photovoice process included photos with captions from the medical school participants, written reflections, and video recorded interviews. Rather than being directive and leading, the lead researcher (LA) used guiding and coaching skills to implement three interactive photovoice sessions for community members and medical students. The three community-based sessions did the following (S2 File): (1) covered theories on SDOH; (2) engendered an exchange of memories between the community members and medical students/faculty around places where people grew up and lived; (3) explained the photovoice processes; (4) implemented a tour of the community lead by neighborhood members.; (5) employed a group exchange to review captured photos; (5) executed a discussion on abandoned buildings and the State Code in West Virginia, and (6) implemented a series of one- to- one video recorded interviews on how the experience might impact their practice of medicine.

Photovoice sessions were created and organized by the lead researcher (LA), a local pastor and community leader, and members of the WVU School of Law, Land Use Clinic. Using an open-ended, iterative set of questions based on a review of literature covering gentrification, place and health theories, and human geography concepts, recorded interviews were conducted by the lead researcher who had over twenty-five years of applied and research experience conducting focus groups, in-depth interviews, and oral histories. At the close of the project all participants, with the addition of WVU SOM medical faculty were involved in at least one of three public exhibitions (S1 File).

## Qualitative data analysis

A qualitative approach was adopted to pick up nuanced experiences and more clearly understand the knowledge, attitudes, and beliefs of the medical students [56–58]. Between June and September 2018, all materials generated by the medical students were analyzed according to the principles of directed content analysis using the SCA framework [59–63].

In order to replicate interrater reliability, increase the credibility of the study findings, mitigate researcher bias in the data, and diminish the likelihood of misinterpretation, methods of triangulation were used [64]. This included review of collected material on at least three occasions during public exhibitions and community sessions by West Side community members along with medical faculty and students (See S1 and S2 Files). Paper drafts were reviewed by the geographer responsible for maps and medical school faculty for agreement. Discrepancies were identified, discussed, and resolved.

## Results

Illustrative quotes from the medical students are used to demonstrate how the photovoice project reflected the SCA framework and place-health model. Additional photovoice data including photos, captions, narratives, and video recorded interviews from the medical students may be found in S4 File.

### 1st level of proficiency

Level 1 proficiency requires the practitioner to avoid a narrow diagnosis of a health problem by connecting health problems to the SDOH domains in the SCA framework which is located around the distribution of resources in the place-health model. Knowledge of the place where that patient resides would help the practitioner to connect that problem to the place and not the individual. Medical students engaged in a lecture on the SDOH, a presentation on the socioeconomic history of the community by a local leader, an interactive discussion between students and community members on experiences growing up in their community, followed by a tour of the neighborhood led by members from the community. At level 1 proficiency the medical students' photovoice data exhibited an understanding of several structural competency domains including: social networks, financial security, residence, risk environments, food access, and education observing a relationship between these factors and the health of the West Side population.

#### Third year male medical student

With all of these adversities, it is not surprising that the West [omit of] Side has experienced worse population health outcomes than other communities in the Charleston area. . .one of the primary barriers to good health on the West Side is the lack of access to various services and resources. Whether it be healthcare facilities, parks, fitness centers, or grocery stores, the distance or the manner in which one must travel in order to reach these locations poses a significant problem to many inhabitants. Poverty is another factor that influences the health of the area. Families may not be able to afford the drugs or services a love[d] one requires.

Extending beyond the direct healthcare effects, it also limits what choices a person may have. . .My involvement in this project has led me to think more about the struggles and circumstances of the patients I encounter in the hospital and clinic. I have more appreciation for what they must endure and overcome, and I believe that this opportunity will serve as a reminder going forward that there is more to healthcare than what lies inside a clinic or hospital.

### 2nd level of proficiency

Level 2 proficiency occurs as the practitioner has knowledge of resources that help patients address the structural competency domains that contravene heath care efforts to improve health. There was an absence of photos, captions, and narratives describing what could be done to address problems of West Side community members. This lack of photovoice data suggested that the medical students had little knowledge about the nonmedical resources available to help West Side community members. The students may have also failed to grasp the nature of the potential structural problems that restrict the ability of healthcare to improve health outcomes. For example, the West Side is heavily populated by abandoned buildings. The students were intensely curious about the problem. To aide them in considering how to

solve a problem of neighborhood blight a discussion on the topic was held with the WVU Law School Land Use Clinic.

## 3<sup>rd</sup> level of proficiency

The 3rd level of structural competency proficiency is depicted as an upstream intervention if Fig 1. Level 3 proficiency is evidenced by the practitioner's ability to see individual level stigmatization within themselves followed by understanding how that becomes structural stigmatization. Level 3 is demonstrated when the practitioner exhibits some understanding of hierarchies and broader sets of power relationships.

Many of the insights from students reflected observations about personal biases and stigmatization. It was harder to find evidence that the students understood how their personal biases could be multiplied inside systems, structures and institutions resulting in regulations, practices, and policies that restrained opportunities and resources for those who have been stigmatized at a personal level.

Several students spoke of what they heard about the West Side from others. One third year medical student explained how he reconciled what he heard about the community with observations from touring the community. The student was able to separate out the stigma and find his own lens with which to view the conditions and members of the community.

One student provided an account of how stories they heard about the West Side impacted their views on the community. It was common for the medical students to admit to never visiting the West Side despite having grown up near the community.

### Third year male medical student

First, the most important thing we as healthcare providers need to do, is recognize and set aside our own personal biases regarding low-income areas. Too many physicians, as I did growing up, simply ignore these areas and do whatever possible to "tune out" the obvious disparities that exist with the residents who live there. This is absolutely unacceptable; if we want change to occur we have to realize and accept the differences and know that, as members of the healthcare community who devote our lives to assisting those in need, it is 100% our responsibility and duty to care for these people, even if they don't reach out to us first.

## 4<sup>th</sup> level of proficiency

Level 4 proficiency, located in Fig 1 at the systems, institutions, and policy level, occurs when practitioners transform into thought leaders and civic members actively contributing to deliberations about the systems, policies, and institutions that play a role in shaping the physical, social, and economic aspects of places. The practitioner should be able to observe and make informed inferences about the interconnectedness of groups to place through social processes, hierarchies and broader sets of power relationships. Photos, captions, and reflections by students demonstrated an awareness of the role of the educational system and economic development in shaping the health and wellness of West Side residents. Data reflecting what it would take to make systemic and policy level changes varied with some placing responsibility on community members and others asserting the need for partnerships or assistance from decision makers in public and private sectors. The analysis of the data failed to demonstrate that the students understood their civic role in addressing the structural issues either as community members or medical practitioners.

Several students saw economic development as a key systems level variable in efforts to improve the lives of West Side community members. A 4th year medical student documented

the need for capital and human development on the West Side. Another 3rd year student demonstrated an awareness of how access to opportunities and resources shaped the lives of West Side residents. This medical student saw education as a key systems level factor that could improve the lives of West Side community members.

**Third year female medical student**

As we drove through the neighborhoods, I noticed the small yards, scarce playgrounds and parks, and no swimming pools. I listened as the poor testing scores of Mary C. Snow West Side Elementary School were discussed, the crime rate of certain areas were noted, and the scarcity of a two-parent household was explained. I also saw families out walking and even caught a glimpse of a few neighborhood kids playing basketball and football in the street on another visit to the West Side. I couldn't help but wonder about the dinnertime conversations that took place in the houses we passed, about the thoughts that played through the minds of young parents and children at night.

## Discussion

Physicians are rarely prepared to consider social and systemic factors because health models and medical training and practices either omit or fail to integrate place-health concepts that could explicate the conditions of places where patients reside. To determine if medical students could understand and incorporate place-health concepts in their practice of medicine a place- health model integrated with a SCA framework was developed and applied to the data generated from a community-based photovoice project. The place-health model integrated into the SCA framework with nine domains, and four levels of proficiency presented a set of metrics useful for evaluating how a community based photovoice project might shift the physician's gaze from the body to the body in a place [65, 66]. The analysis of qualitative data using the place-health model and SCA framework revealed the following.

First, the medical students were able to use the community-based photovoice process to understand a connection between place and health. Use of the place-health model and the SCA framework to analyze the photovoice data demonstrated a shift in the student's discourse and narratives that described how they conceptualized health beyond the individual body as the reference point. The students were able to work at level 1 proficiency which is located in the place-health model around the SDOH (Fig 1).

The student's photovoice data demonstrated very little knowledge of how to address structural, systemic, and policy-based problems as specified by level 2 proficiency. Similarly, at level 3 proficiency the student's photovoice data reflected individual level stigma learned from family and other relationships but not how the individual level stigmatization multiplied from the individual into the structure of institutions. While the photos, captions, video interviews, and narratives of students mentioned having biased opinions on the West Side without ever having visited, photovoice data, interviews, and narratives showed no evidence of thoughts about how the formation of those biases could be part of the larger decision-making systems of the City of Charleston or the healthcare systems of which they were members. Understanding hierarchical structures and power-based relationships, depicted as an upstream intervention in Fig 1, was not reflected in the photovoice data which would have demonstrated the basis for achieving level 3 proficiency.

Last, the photovoice data of the students did not demonstrate achievement of level 4 proficiency where they were meant to display evidence of a merger between their physician-self and a civic or political role. The photos and narratives did not show that the students understood

how to utilize their knowledge, values, and standing on behalf of policies or actions that address and improve the structural issues confronting the West Side community where their patients lived.

Using the nine domains of structural competency which are located around the SDOH in Fig 1, the analysis of the photovoice data demonstrated that six of nine structural competency domains were identified by the medical students including: financial security, residence, risk environments, food access, social networks, and education. Alternatively, the results of the analysis indicate that the photovoice exercise did not achieve an understanding of the domains of discrimination, legal status, and structural stigmatization by the medical students.

There are several limitations to this study that might explain why the analysis failed to find evidence that the students did not reach structural competency proficiency beyond level 1 and did not display awareness of specific domains in the photovoice data. It is possible that the photovoice exercise is of limited ability to portray the more complicated ideas of the place-health model and the SCA framework. Of equal importance as a limitation is that the photo-voice exercise did not integrate other opportunities to learn about historical factors including the community and the socioeconomic changes that shaped the West Side over time. The medical students were engaged in a discussion with community members, a tour of the neigh-borhood lead by community representatives, and a historical, socioeconomic overview by a local pastor and leader of a church and community development corporation. However, these photovoice elements alone might not have been sufficient to give the students a full under-standing of structural, institutional, and systems issues that make up a place.

The historical and present-day discrimination, power imbalances, and structural stigmati-zation may not have shown up in the qualitative data from the medical students because these are complex, emotion-charged topics that make them harder to identify, discuss, or admit. In the case of this structural competency framework medical students would be called upon to make the connection between the present-day existence of stigma and the use of the stereo-types to disadvantage groups in a range of socioeconomic opportunities including housing, healthcare, employment, educational opportunities etc. [17, 67, 68]. That discrimination is generally rooted in history, still lingers in the present, and requires societies to be responsible for hard-to-define corrective acts makes it all the more difficult to discuss as the notion of a color blind society takes hold making it difficult to make the case for current day discrimina-tion. [68, 69].

A final limitation to this study may be the validity and reliability of the SCA framework as defined and operationalized, The concept of structural competency is in its infancy and addi-tional work may be needed beyond a qualitative analysis of participants' photovoice data including quantifying and evaluating outcomes by external measures [44, 70].

## Conclusion

Undoubtedly research confirms that health is the result of the relationship between bodies and places and the structural and social factors that emanate from that place. This study con-structed an integrated place-health model and structural competency analytical framework to analyze qualitative data from a community based photovoice exercise that immersed medical students in a place where many of their patients lived.

The place-health model and SCA framework presented a set of metrics by which to evaluate efforts to shift the physician's gaze to the body in a place. When analyzed, the photovoice data demonstrated that medical students achieved structural competency at level 1 proficiency and came to understand how health might be connected to place represented by six of the nine domains of the SCA framework.

Given the importance of place-health theories to explain population health outcomes, interventions, curricula, and programs based on a place-health model and structural competency analytical framework could help medical students merge the image of patients as singular bodies into bodies set within a context. Our findings suggest that medical student's engagement with place-health systemic, institutional and structural forces deepens when they co-create narratives of their lived experiences in a place with patients as community members during a community-based photovoice project.

## Supporting information

**S1 File. Exhibitions, video, and GIS maps.**
(DOCX)

**S2 File. Community sessions.**
(DOCX)

**S3 File. Adaption of structural competency framework.**
(DOCX)

**S4 File. Photovoice medical students.**
(DOCX)

## Author Contributions

**Conceptualization:** Lauri Andress.

**Data curation:** Lauri Andress.

**Formal analysis:** Lauri Andress, Matthew P. Purtill.

**Methodology:** Lauri Andress, Matthew P. Purtill.

**Project administration:** Lauri Andress.

**Supervision:** Lauri Andress.

**Writing – original draft:** Lauri Andress, Matthew P. Purtill.

**Writing – review & editing:** Matthew P. Purtill.

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
