## [Decision Letter · Decision Letter 0]

5 Dec 2019

PONE-D-19-22363

Shifting the Gaze of the Physician from the Body to the Body in a Place: A Qualitative Analysis of a Community-based Photovoice Approach to Teaching Health-Place Concepts to Medical Students

PLOS ONE

Dear Dr. Andress,

Thank you for submitting your manuscript to PLOS ONE. After careful consideration, we feel that it has merit but does not fully meet PLOS ONE’s publication criteria as it currently stands. Therefore, we invite you to submit a revised version of the manuscript that addresses the points raised during the review process.

The two reviewers have presented thorough comments on the manuscript to provide guidance in revisions. I concur with their recommendations. Please review these and address each point. I look forward to reading your revised version.

We would appreciate receiving your revised manuscript by Jan 19 2020 11:59PM. To enhance the reproducibility of your results, we recommend that if applicable you deposit your laboratory protocols in protocols.io, where a protocol can be assigned its own identifier (DOI) such that it can be cited independently in the future. For instructions see: http://journals.plos.org/plosone/s/submission-guidelines#loc-laboratory-protocols

We look forward to receiving your revised manuscript.

Kind regards,

Heidi H Ewen

Academic Editor

PLOS ONE

Journal Requirements:

2. Please provide additional details regarding participant consent. In the ethics statement in the Methods and online submission information, please ensure that you have specified (1) whether consent was informed and (2) what type you obtained (for instance, written or verbal). If your study included minors, state whether you obtained consent from parents or guardians. If the need for consent was waived by the ethics committee, please include this information.

3. When reporting the results of qualitative research, we suggest consulting the COREQ guidelines: http://intqhc.oxfordjournals.org/content/19/6/349. In this case, please consider including more information on the number of interviewers, their training and characteristics; participants characteristics (age, years of study, etc); how data were collected and analysed. Moreover, we suggest that the quotations are identified (e.g., with a participant number); and as we note that the video link included does not work, we suggest removing it.

4. Please include your tables as part of your main manuscript and remove the individual files. Please note that supplementary tables (should remain/ be uploaded) as separate "supporting information" files

Reviewers' comments:

Reviewer's Responses to Questions

**Comments to the Author**

1. Is the manuscript technically sound, and do the data support the conclusions?

Reviewer #1: Yes

Reviewer #2: Yes

2. Has the statistical analysis been performed appropriately and rigorously? 

Reviewer #1: Yes

Reviewer #2: N/A

3. Have the authors made all data underlying the findings in their manuscript fully available?

Reviewer #1: Yes

Reviewer #2: Yes

4. Is the manuscript presented in an intelligible fashion and written in standard English?

Reviewer #1: No

Reviewer #2: Yes

5. Review Comments to the Author

Reviewer #1: This is brilliant and very important work but it should be re-written so that the points the authors are trying to make are stated more clearly. The current version tries to convey much too much information, and too many ideas. There is an overabundance of jargon and technical language and almost total lack of discussion of how photography was used or could be used. In addition, it talks about photovoice but does not explain what photovoice is, nor does the body of the article present or discuss specific photographs. The concepts in this paper are very, very important and the authors' approach is groundbreaking and potentially transformative for medical education and practice. I would strongly recommend that this article be accepted with the agreement that the authors work on clarity and paring it down to identify and support the most important points.

Reviewer #2: Physician pv manuscript comments:

• The title refers to "Health-Place" but place-health is used throughout, along with other combinations of the words place and health. Need to be consistent or explain why one is unique to the other.

• Citizens instead of citizen on line 52

• (In the area of lines 82-90) A consideration for the authors is to think about the language and training of the medical community. Just as a physician diagnoses the symptomology of the body for diagnosis and direction of a treatment plan (Individual as patient) the same diagnostic process should be seen as viable for assessing the Community as patient when diagnosing the symptomology of the community’s health which can point to potential treatment plans, albeit at a systemic level. A key caveat in this statement is to caution the authors to use this opportunity to educate the health care industry that they should be partnering with public health and NOT feeling the need to be physician of the body AND the physician, if you will, of the physical/social community.

• On lines 95-97 the authors state” “The presence of these downstream health problems suggests that causation and solutions to health problems go beyond individual bodies defined as patients to incorporate a set of systemic institutional, social, and structural issues to bodies now defined as social beings”. The beginning of the sentence is fine but then goes off the rails when it concludes with “…to incorporate a set of systemic institutional, social, and structural issues to bodies now defined as social beings”. Now defined as social beings? Please consider rethinking and rephrasing.

• How are bodies firstly groups? (lines 101-102).

• The authors use the abstract phrasing of ‘cultural toolkit’ without much explanation. Either rephrase or take the time to explain the use, or the reason for the use, of the phrase. (line 108) What is the role of cultural competence in the cultural toolkit? How does cultural competence contribute to or modify the SDOH and structural competence? (lines 107-108)

• (line 113) ‘convert spaces into places’ is to abstract for a more pragmatic physician audience. Please consider the rethinking and rephrasing.

• A general observation regarding the comments above. Some of the language is likely accessible to the already converted but unsure if it might be a bit too distally worded to resonate with those stuck in a biology based mental frame as many in the health care sector are. A few direct and less flowery statements would help to jar the more resistant readers to new ways of thinking.

• Cultural competency is: "A set of behaviors, policies, and attitudes which form a system or agency which allows cross-cultural groups to effectively work professionally in situations" and "the integration and transformation of knowledge about individuals and groups of people into specific standards, policies, practices, and attitudes used in appropriate cultural settings to increase the quality of services; thereby producing better outcomes." (lines 140- ) Rethink and rephrase.

• Consider rethinking and rewording lines 137-139 to make more direct and clear; same with lines 150-151.

• Can structural competence totally account for this? What is the role of cultural competence in understanding the connection? (lines 179-181)

• What was the racial and ethnic makeup of participants? Were they native to Charleston or from elsewhere? What was their socioeconomic background? How did these factors influence their cultural competence in understanding the West Side community?

• Photovoice not ‘photovoie’ (line 241)

• Practitioners not ‘practitioner’ (line 242)

• (lines 234-238 but also in other areas of manuscript) Photovoice’s ability to highlight the values and priorities of community members (emic) as opposed to those of the researchers or other outside groups (etic) is a foundational strength of the method, and yet, in this application of photovoice the authors largely chose to retain ‘power’ with the etic/outsider medical community residents to reflect the lived realities of other people. In this way the author’s application is somewhat antithetical to the method’s intended roots. While one can see the author’s aims in their matching of physician residents with community members one is left to wonder why they did not rather choose to let the community members represent their lives themselves through photography, discussion, captioning and public display? At one level this is a fatal flaw of the design. At the same time the authors findings do contribute to the generation of new knowledge for advancing strategies for engendering empathy within the medical profession and the training therein. It would be nice for the authors to at a minimum recognize this limitation of the study being presented and to reflect on an alternative approach that honors and values the power and voice of those so frequently oppressed. Physician residents can still be a part of the process, but a serious balancing of power should be part of the equation.

• (line 257) photovoice not ‘photo voice’.

• West Side not ‘West of Side’ (line 298)

• Loved one not ‘love one’ (line 304)

• Why is the work ‘Proficiency’ in the middle of line 328?

• Students’ not student’s (line 331)

• Because this statement is provided on line 341 “When asked what they knew about the West Side students admitted to never visiting the West Side despite having grown up near the community.” It would seem appropriate to elaborate the thought for the reader. One would assume literature discusses exposure and experience to a growing sense of awareness and potential to budding cultural humility, but it would be good for the authors to elaborate on the point rather than expecting the reader to assume the reason for the statement.

• On lines 360-362 the authors state: “The analysis of the data failed to demonstrate that the students understood their civic role in addressing the structural issues either as community members or medical practitioners.” But as a reviewer I am left with the question as to why the authors believe this? Or what their speculative reasoning might be. As a reader one can conclude that they are simply unaware of the duty as seen by the authors, on the other hand the nature of the task seems more aligned with exposure, awareness, and empathy with little discussion of professional role, a larger civic duty or educational challenge to socially structured mores. They do discuss this to a degree in the later part of the paper, but to implicate the methods failure to elicit this action-oriented state may say more about the implementation of the method than anything. Interventions, in this case a unique application of a photovoice-type intervention, can only be expected to measure outcomes that are targeted as process activities of the intervention. To the extent that additional outcomes are a result is interesting but should not be necessarily expected.

• Capital not ‘capitol’ (line 365).

• On line 460 the authors state: “The medical student’s photovoice data showed an achievement of structural competency at proficiency at level 1 with a failure to achieve proficiency at levels higher than that.” Similar to the earlier comment about the nature of the intervention design, it is challenging to fully expect changes at higher levels without commensurate intervention activities to achieve those aims. Is it possible that incorporating cultural competency might contribute to improving proficiencies at a higher level? (also the ‘than that’ is not needed at the end of the sentence).

• As a researcher who values graphics to convey complex thinking and processes this reviewer appreciates the author’s attempt to convey their ‘Production of Inequities Cultural Toolkit’; however, the organization and flow of the diagram needs a serious rethinking if it is to convey the authors intent. A consideration might be to position the elements within the image into a social ecological frame. As drawn the reader is unclear as to how to process the information provided, where the beginning or end of the model is for the reader, or which elements are grouping labels, and which are constructs. There is too much to unpack to directly rebuild the model, but consideration might be given to enlisting the insights of a fellow researcher who is gifted at graphically displaying complex relationships.

6. PLOS authors have the option to publish the peer review history of their article (what does this mean?). If published, this will include your full peer review and any attached files.

Reviewer #1: Yes: Mark Rosenberg

Reviewer #2: No

---

## [Author Response · Author response to Decision Letter 0]

1 Jan 2020

I have responded to the reviewer's comments and as well as comments from the editorial staff by uploading a file titled response to reviewers.

---

## [Decision Letter · Decision Letter 1]

22 Jan 2020

Shifting the Gaze of the Physician from the Body to the Body in a Place: A Qualitative Analysis of a Community-based Photovoice Approach to Teaching Place- Health Concepts to Medical Students

PONE-D-19-22363R1

Dear Dr. Andress,

We are pleased to inform you that your manuscript has been judged scientifically suitable for publication and will be formally accepted for publication once it complies with all outstanding technical requirements.

With kind regards,

Heidi H Ewen

Academic Editor

PLOS ONE

Additional Editor Comments (optional):

Reviewers' comments:

Reviewer's Responses to Questions

**Comments to the Author**

1. If the authors have adequately addressed your comments raised in a previous round of review and you feel that this manuscript is now acceptable for publication, you may indicate that here to bypass the “Comments to the Author” section, enter your conflict of interest statement in the “Confidential to Editor” section, and submit your "Accept" recommendation.

Reviewer #1: All comments have been addressed

2. Is the manuscript technically sound, and do the data support the conclusions?

Reviewer #1: Yes

3. Has the statistical analysis been performed appropriately and rigorously? 

Reviewer #1: Yes

4. Have the authors made all data underlying the findings in their manuscript fully available?

Reviewer #1: Yes

5. Is the manuscript presented in an intelligible fashion and written in standard English?

Reviewer #1: Yes

6. Review Comments to the Author

Reviewer #1: the paper is significantly improved. It needs some proofing and copyediting, but these are minor. Some small suggestions: p3 line 96: the word should probably be systems "theory" rather than "systems therapy." p15, line 465-467--this is confusing. delete the words "the analysis failed to find evidence that" and let the rest of the sentence stand. Reviewer 2 made very good suggestions and raised important questions. The changes made in response to reviewer 2's comments really improved the paper.

7. PLOS authors have the option to publish the peer review history of their article (what does this mean?). If published, this will include your full peer review and any attached files.

Reviewer #1: Yes: Mark L. Rosenberg, MD, MPP

---

## [Editor Report · Acceptance letter]

4 Feb 2020

PONE-D-19-22363R1 

Shifting the gaze of the physician from the body to the body in a place: A qualitative analysis of a community-based photovoice approach to teaching place-health concepts to medical students 

Dear Dr. Andress:

I am pleased to inform you that your manuscript has been deemed suitable for publication in PLOS ONE. Congratulations! Your manuscript is now with our production department. 

With kind regards,

on behalf of

Dr. Heidi H Ewen 

Academic Editor

PLOS ONE